# The Effects of Body Composition Characteristics on the Functional Disability in Patients with Degenerative Lumbar Spinal Stenosis

**DOI:** 10.3390/jcm12020612

**Published:** 2023-01-12

**Authors:** Yesull Kim, Chanhong Lee, Hyunji Oh, Ji-Seon Son, Aram Doo

**Affiliations:** 1Department of Anesthesiology and Pain Medicine, Jeonbuk National University Medical School and Hospital, Jeonju 54907, Republic of Korea; 2Research Institute of Clinical Medicine of Jeonbuk National University-Biomedical Research Institute of Jeonbuk National University Hospital, Jeonju 54907, Republic of Korea

**Keywords:** body composition, fat, function, low back pain, obesity

## Abstract

Several research studies suggest that obese patients are at a higher risk of developing lumbar spinal disorder, including degenerative lumbar spinal stenosis (LSS), compared to normal-weight individuals. However, there are few investigations of how obesity affects functional disability in activities of daily living (ADL) in patients who were diagnosed with LSS. This prospective observational study aimed to determine if an association exists between body composition parameters, such as body fat and skeletal muscle, and functional disability in ADL of LSS patients. In the results of the current study, there were significant differences in percent body fat between the mild/moderate and severe disability groups. However, there were no differences in skeletal muscle mass or index between the two groups. Furthermore, we found a positive linear relationship between percent body fat and functional disability in male sex. This study suggests that increased percent body fat predicts potential severe functional disability in ADL in LSS patients. Body composition analysis may provide useful information for predicting the disease severity of various lumbar spinal disorders in clinical practice.

## 1. Introduction

Spinal pain is a growing concern in aging populations with increasing life expectancy worldwide. Degenerative lumbar spinal stenosis (LSS) frequently causes typical symptoms, including neurologic intermittent claudication, sciatica and low back pain. In addition, vague leg symptoms, such as paresthesia, weakness and altered sensation, may exist [1,2]. These characteristic manifestations may lead to functional disabilities in activities of daily living (ADL), decreased quality of life and increased social isolation, especially in older adults. 

Functional disability in ADL may be a representative of the disease severity of LSS, because it means how the spinal pain and neurologic deficit affects the patients’ daily life. It has been reported that the greater the intensity of symptoms, the worse the functional disability [3]. Furthermore, evaluating functional disability in ADL may help to establish the treatment plan and even predict the post-surgical outcome for patients with LSS [4]. Radiologic imaging, such as magnetic resonance imaging (MRI), is used to decide the severity of LSS with more clarity, but it is greatly limited. It is well known that the correlation between the functional status and the severity of radiologic findings is often unmatched [5,6]. Despite the importance of functional disability in ADL in LSS patients, there are no clear predictive factors associated with it.

Obesity has been increasingly investigated as a potential predictor for the development of various lumbar spinal disorders, but the results are inconsistent, and the causal relationships are also unclear [7,8,9,10,11,12,13]. Several researches suggested that obese patients were at a higher risk of developing lumbar spinal disorder, including LSS, compared to normal-weight individuals [10,11]. Other studies reported that obesity was associated with the severity of the lumbar intervertebral disc degeneration and even with the less functional improvement after surgery in LSS patients [8,14,15]. However, there are few investigations of how obesity (increased body fat) affects functional disability in ADL in patients who were diagnosed with LSS.

The aim of the current study was to determine if an association exists between body composition characteristics and functional disability in ADL of LSS patients, especially to understand which body fat parameters could accurately predict it. We further expected that other body composition parameters related to the amount of skeletal muscle also affect it. Additionally, the therapeutic effects of epidural steroid injection of LSS patients were evaluated.

## 2. Materials and Methods

This prospective observational study was approved by the Institutional Review Board of the author’s hospital and registered with the WHO International Clinical Trials Registry Platform (KCT0006046). This manuscript adheres to the applicable STROBE guidelines. After obtaining written informed consent, a total of 72 consecutive patients who visited our pain clinic in a tertiary care hospital due to LSS were enrolled in the study. Inclusion criteria were as follows: (1) age >40 years; (2) clinical LSS symptoms (radiculopathy +/− low back pain) more than 3 months; (3) symptom intensity with numeric rating scale (NRS; 0–10) of 4 or more; (4) radiologic confirmation of LSS through magnetic resonance imaging (MRI). Patients with a literacy problem or language difficulties; a history of psychotic disorder or drug abuse; chronic opioid usage over 3 months; a concomitantly complicated spinal disease, including epidural lipomatosis; ligament ossification or diffuse idiopathic skeletal hyperostosis; a definite indication for prompt surgery, such as cauda equina syndrome; a history of previous spinal surgery; orthopedic metal implants in any body region; and cardiac pacemaker or implantable cardioverter-defibrillator (ICD) were excluded from the study.

### 2.1. Evaluation of Functional Disability in ADL and Group Allocation

The functional disability in ADL was evaluated using a validated Korean version of the Oswestry Disability Index (ODI) [16]. All participants were asked to complete the self-reported ODI questionnaire in their first visit to our department. ODI is a widely used, self-reported questionnaire for assessing a patient’s functional disability in ADL in patients with lumbar spinal pain. ODI consists of 10 items that are disease-associated health status measurements of how pain affects functional disability in ADL in patients with lumbar spinal pain. Each item is rated on a 6-point scale that ranges from 0 to 5, and the global score is added and multiplied by 2. Therefore, the global score ranges from 0 to 100. The higher the global score, the greater the functional disability in ADL. We divided participants into two groups, the mild to moderate disability group (ODI ≤ 40) and the severe disability group (ODI > 40), as previous studies suggested [17].

### 2.2. Body Composition Analysis Using Bioelectrical Impedance Analysis (BIA)

All participants underwent body composition measurement using a body composition analyzer (Inbody S10^®^, Biospace, Seoul, Republic of Korea), according to the manufacturer’s guidelines, at their first visits to our department. Inbody S10^®^ provides various information for body composition through BIA methods. Patients were asked to take a rest in the supine position on a nonconductive table for at least 10 min before measurements. The skin surface where the electrode would be placed was cleansed with an alcohol swab before application, in keeping with standard practice. Then, the surface electrodes were attached to both fingers (thumb and middle finger) and ankles, according to the manufacturer’s instructions. Body composition parameters, such as body fat mass (kg), percent body fat (%), skeletal muscle mass (kg) and skeletal muscle index (kg/m^2^), were recorded.

### 2.3. Outcome Measures

Other demographic factors that may be associated with functional disability in ADL, including age, sex, comorbid disease, radiologic severity of LSS (degree of stenosis and number of stenotic segment) and duration of symptoms, were also evaluated. The degree of spinal canal stenosis was categorized from A to D, according to the previous method of classification by Schizas et al. [18] using MRI. That of foraminal stenosis was categorized as mild, moderate or severe stenosis. 

We also measured the cross-sectional area (CSA) of the psoas and multifidus muscles, using T1-weighted axial MRI images at the midpoint of the L3/4 intervertebral disc. Using picture archiving and communication system imaging measurement software (Infinitt healthcare, Seoul, Republic of Korea), the CSA of the multifidus and the psoas muscles were measured, as presented in Figure 1. All measurements were independently analyzed by two pain physicians who were blinded to patients’ characteristics, and a consensus had to be found. In case of disagreement with a difference of more than 15%, images were reviewed together and discussed until the consensus was gained.

### 2.4. Epidural Steroid Injection Procedure

All participants received a target-specific transforaminal epidural steroid injection in the involved spinal segment judged from the patients’ symptoms and MRI findings at their first visits. All procedures were performed with guidance with a fluoroscope (ARCADIS Orbic^®^; Siemens AG, Erlangen, Germany) by an experienced pain physician. The patients were placed in a prone position, and standard sterile preparation was conducted. Using the ipsilateral oblique view on the fluoroscopy to obtain a traditional “safety triangle” image [19], a 22-gauge spinal needle was introduced just inferior to the pedicle using the tunnel vision technique. After reaching the intervertebral foramen, the final position of the needle tip was adjusted using the anteroposterior and lateral fluoroscopic view. After confirming appropriate epidural contrast uptake, 5 mL of 0.6 % lidocaine containing 5 mg dexamethasone was administered. For the multi-level procedure at two or more sites, 5 mL of drug was used at each level, with a divided dose of dexamethasone not to exceed a total dose of 5 mg. The number of epidural blocks during 4 weeks and the degree of pain relief using the changes of NRS at the first visit and 4-weeks follow-up were recorded.

The body composition parameters, including body fat mass (kg), percent body fat (%), skeletal muscle mass (kg), skeletal muscle index (kg/m^2^) and other demographic characteristics, were compared between the mild/moderate and severe disability groups. The primary outcomes were the differences of the body composition parameters between the two groups. Secondary outcomes included epidural block-related therapeutic effects in the two groups. 

### 2.5. Sample Size Determination and Statistical Analysis

Sample size was predetermined by the *t*-test sample size calculation, using IBM SPSS Statistics for Windows, version 27, based on the assumption that the minimum detectable difference between the two groups in percent body fat was 10%. A total of 64 patients were required, with a significance level of 0.05 (α = 0.05) and a power of 80% (β = 0.20). Considering the predicted dropout rate, the total sample size was increased to 72.

All descriptive data are expressed as the number of patients, mean ± SD, and median [interquartile range]. Continuous variables were analyzed by a two-tailed *t*-test or Mann—Whitney rank-sum test after a normality test. The χ^2^ test was used to compare categorical variables. Linear regression analysis was performed to verify the relationships between body composition parameters and the ODI scores. A *p*-value of less than 0.05 was considered statistically significant. All statistical analyses were performed using IBM SPSS Statistics for Windows, version 27.

## 3. Results

### 3.1. Study Participants and Patient Characteristics

Among the 72 participants enrolled, 63 patients (29 in the mild/moderate group and 34 in the severe disability group) completed the study. The CONSORT flow diagram is shown in Figure 2. The average ODI score was 26.9 and 55.6 in the mild/moderate and severe disability groups, respectively (*p* < 0.001). Patient demographics, including age, sex, body weight, body mass index (BMI) and radiologic severity of LSS, were compared between the two groups. No significant differences of these parameters were observed between the two groups (Table 1).

### 3.2. Study Outcomes

Body composition parameters and cross-sectional areas of the spinal muscles of the two groups are shown in Table 2. Percent body fat (%) was significantly higher in the severe disability group compared to the mild/moderate group in each sex (*p* = 0.016 in male and *p* = 0.004 in female). The proportion of obese patients (percent body fat ≥17% in men and ≥32 % in women) was compared, and it was significantly higher in the severe disability group compared to the mild/moderate group only in females (*p* = 0.028 by chi-square test).

The linear regression analysis revealed that percent body fat was positively correlated to the ODI score in males (r = 0.554 and *p* = 0.005). However, there was no statistically significant relationship between them in females (r = 0.331 and *p* = 0.052) (Figure 3).

Meanwhile, there were no significant differences in the body composition parameters related to skeletal muscle, including skeletal muscle mass (kg) and skeletal muscle index (kg/m^2^), between the two groups. However, the CSA of the spinal muscles, such as psoas and multifidus, were significantly different between the two groups in each sex. The CSAs of psoas muscles were significantly smaller in the severe disability groups compared to the mild/moderate group in each sex (*p* < 0.001 in male and *p* = 0.020 in female) (Table 2). In further analysis, there were significant linear relationships between the CSA of psoas and multifidus muscles and the ODI score (Figure 4A,B).

The epidural block procedure-related pain-relieving effect is shown in Table 3. The NRS score at 4 weeks following the first epidural block was significantly lower in the mild/moderate disability group compared to the severe disability group (*p* < 0.001). The patients in the severe disability group experienced a smaller pain-relieving effect from the epidural block compared to the mild/moderate disability group (45.1% vs. 25.2%, *p* = 0.002).

## 4. Discussion

The current study demonstrated that LSS patients with severe disability in ADL (ODI score > 40) had a significantly higher fat mass and percent body fat compared to subjects with mild to moderate disability. The higher the percentage of body fat, the greater the functional disability in ADL, with a linear relationship. As previous studies suggested that radiologic imaging is limited for evaluating disease severity [5,6], radiologic severity of LSS was comparable between the mild/moderate and severe disability groups in this study. Meanwhile, LSS patients with severe disability experienced a smaller therapeutic effect from repeated epidural blocks compared to those with mild/moderate disability. This result suggests that evaluating functional disability in LSS patients potentially helps to predict the response to conservative treatment, such as epidural blocks, and to establish more effective treatment options promptly. Many studies suggested that overweight and obesity was positively related to the development of lumbar spinal diseases [10,11,12,13]. However, the precise mechanisms of the association are little known, and the causal relationships are also unclear. Some studies believe that excessive mechanical overload might affect the development of degenerative spinal disease [10,12]. Others suggests that obesity might be the result of the limited daily activity caused by spinal pain [20]. Despite numerous studies, there are few investigations of how obesity affects functional disability in patients who were diagnosed with LSS.

Fanuele et al. revealed that general and spinal disease-associated functional status was significantly worse in overweight and obese patients compared to normal weight individuals in a large population-based cross-sectional survey [21]. However, they declared the possible inaccuracy of self-reported BMI as one of the limitations of the study. BMI is a commonly used tool for clarifying obesity in most studies [8,10,11,14,21]. It is a simple, traditional and easily applicable method to judge body fatness in clinical practice. However, BMI, which is body weight in kilograms divided by height in meters squared, is not the accurate measurement of body fat. It cannot distinguish between body fat and lean body mass nor reflect accurate body fat mass and distribution. BMI might over- or underestimate body fat, and it can be different according to sex, age, ethnicity and individual variations [22,23].

The current study proved that percent body fat was a better predictor for functional disability in LSS patients than BMI was, even though the mechanism of association between percent body fat and functional disability in ADL in LSS patients remains unclear. Although percent body fat was significantly different between the mild/moderate and severe disability group, the BMI was similar in the two groups. Therefore, we suggest that the body composition analysis could be used as a part of a supplementary diagnostic tool for predicting the functional status of LSS patients in clinical practice. BIA is a reliable, inexpensive, fast and bed-side applicable tool for measuring body composition. It allows estimating the composition of various body compartments, such as body fat, fat-free mass, body water or body cell mass in both healthy and ill subjects [24,25,26]. BIA has also been recently applied to evaluate nutritional or metabolic status, as well as body hydration status, in patients with various entities of diseases [27]. BIA might be able to expand its role for predicting the severity or the prognosis of various painful musculoskeletal disorders, and further investigations in a large population should be performed.

The current study also aimed to evaluate whether the severe disability of LSS patients was associated with the skeletal muscle parameters of BIA, such as skeletal muscle mass or index. However, we cannot find those associations. Indeed, sarcopenia has been focused on as associated with decreased function in ADL in older patients, with further association with several musculoskeletal disease entities [28]. Furthermore, the importance of the paraspinal muscle has been investigated in the clinical progress of various musculoskeletal diseases of the lumbar spine. Structural change, such as atrophy or fat infiltration in the multifidus and paraspinal muscles, has been observed in patients with chronic low back pain [28,29,30,31]. In the current study, skeletal muscle parameters of BIA were not beneficial enough for predicting the functional disability of LSS patients, but radiologic measures of psoas and multifidus muscles seem to have the potential to predict it. A future study of a large population should be performed to explain these associations clearly.

These are limitations of this study. First, this study was limited to a relatively small number of participants of a single hospital. Second, the causality of the patients’ LSS and their body composition characteristics were not discussed. Third, the long-term effect of high percent body fat or reduction of body fat on LSS was not evaluated in this study. The effects of physical or surgical weight reduction on the improvement of patients’ symptoms and function are still inconclusive [32,33]. Further observational cohort studies can be conducted.

## 5. Conclusions

In conclusion, this study suggests that increased percent body fat predicts potential severe functional disability in ADL in LSS patients. BIA may provide useful information for predicting the severity or the prognosis of various lumbar spinal disorders in clinical practice.

## Figures and Tables

**Figure 1 jcm-12-00612-f001:**
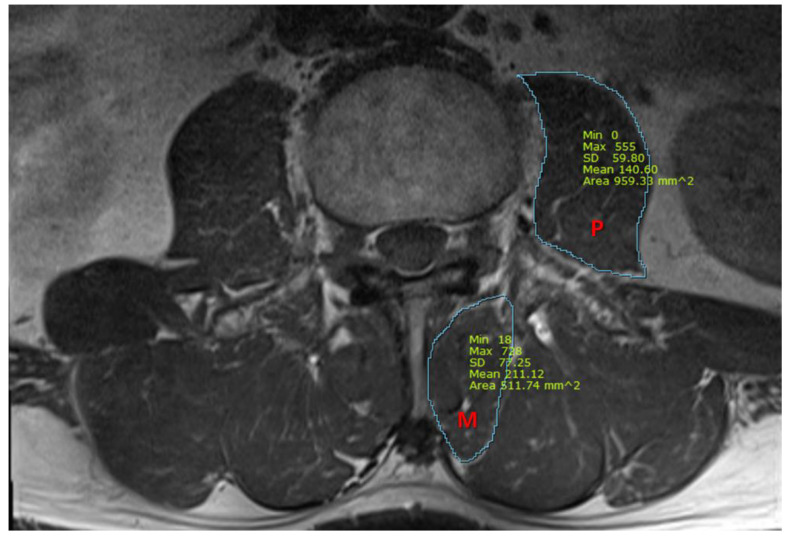
A representative magnetic resonance image for the measurement of cross-sectional area (mm^2^) of the psoas and multifidus muscles. P, psoas muscle; M, multifidus muscle.

**Figure 2 jcm-12-00612-f002:**
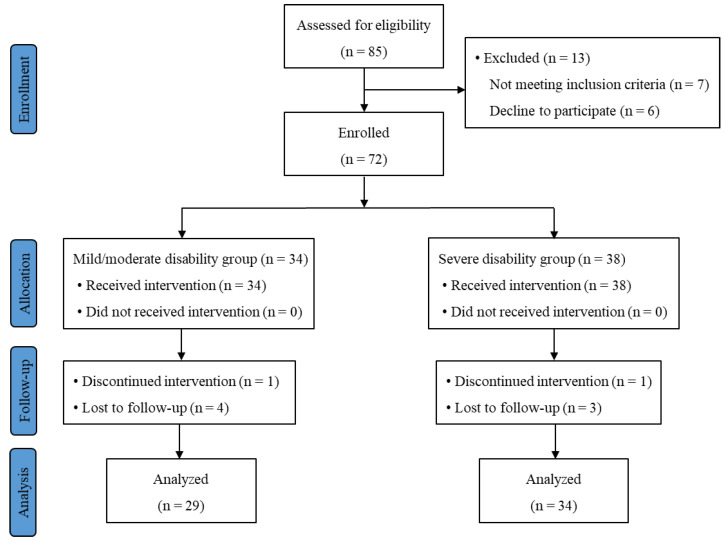
Subject flow diagram.

**Figure 3 jcm-12-00612-f003:**
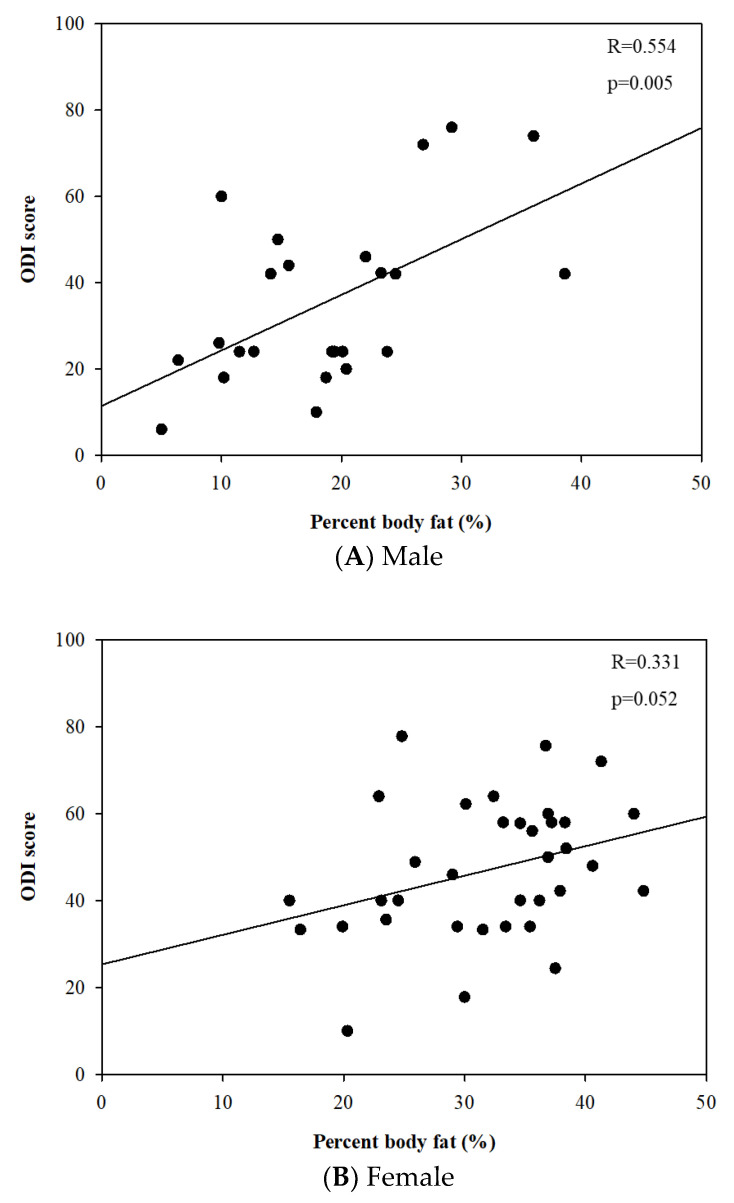
Relationships between percent body fat (%) and Oswestry disability index score.

**Figure 4 jcm-12-00612-f004:**
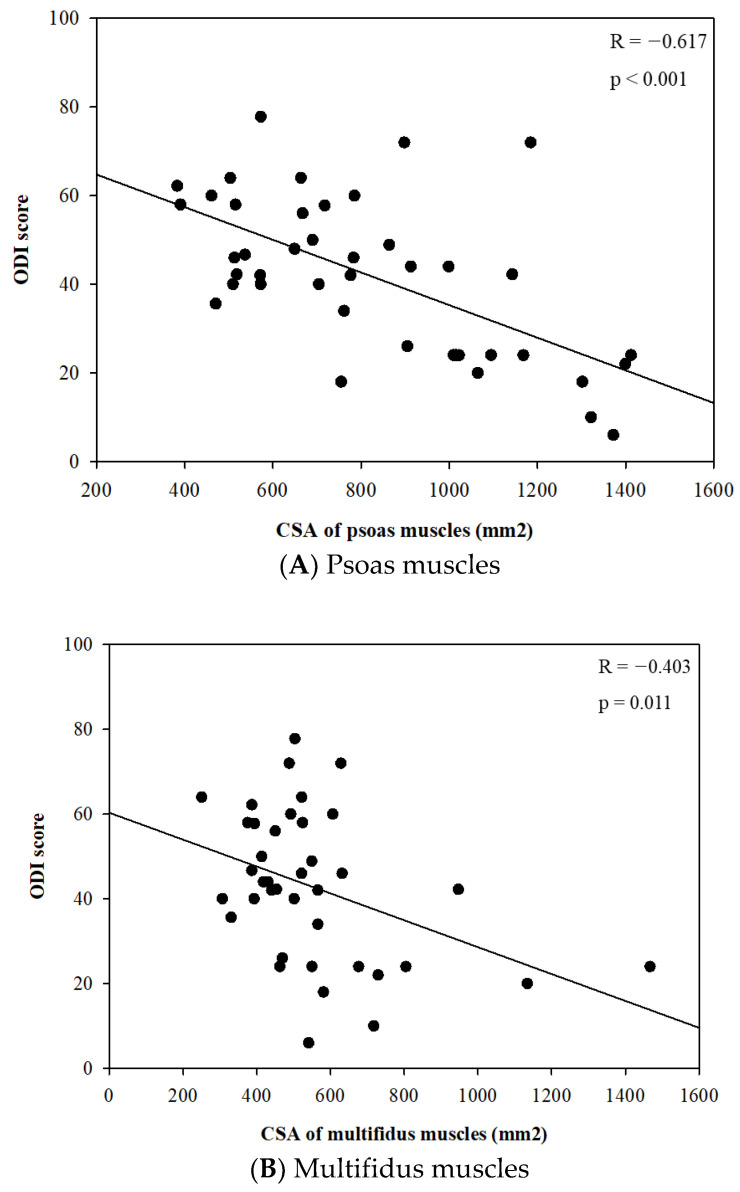
Relationships between cross-sectional area (CSA) of psoas and multifidus muscles (mm^2^) and Oswestry disability index score.

**Table 1 jcm-12-00612-t001:** Patient demographics.

	Mild/Moderate Disability (n = 29)	Severe Disability (n = 34)	*p* Value
Age (years)	68.0 (63.5–73.0)	69.5 (60.0–76.0)	0.879
Sex (M/F)	14/15	11/23	0.198
Body weight (kg)	60.7 ± 10.7	63.6 ± 11.9	0.301
BMI (kg/m^2^)	23.6 ± 2.8	25.1 ± 3.2	0.053
Underlying metabolic diseases			
Hypertension	10 (34.5%)	7 (20.6%)	0.216
Diabetes mellitus	5 (17.2%)	4 (11.8%)	0.721
Hyperlipidemia	4 (13.8%)	3 (8.8%)	0.694
Radiologic severity of LSS			
Degree of spinal canal stenosis			
Grade A/B/C/D	14/3/5/2	8/7/10/2	0.143
Degree of foraminal stenosis			
Mild/Moderate/Severe	4/6/11	0/3/9	0.374
Number of stenotic segment			
1/2/3 or more	1.5 (1.0–2.0)	1.0 (1.0–2.0)	0.605
Duration of symptom (months)	12.0 (5.3–57.0)	6.0 (2.0–27.0)	0.173
Type of analgesic medications			
NSAIDs	7 (24.1%)	10 (29.4 %)	0.638
Acetaminophen/tramadol combinations	4 (13.8%)	9 (26.5%)	0.215
Opioids	1 (3.4%)	2 (5.9%)	1.000
Total ODI score	26.9 ± 9.8	55.6 ± 11.4	<0.001 *

Data are presented as median (interquartile range), numbers, mean ± standard deviations, or numbers (percentages). Mild/moderate disability was defined as Oswestry disability score (ODI) ≤40. Severe disability was ODI > 40. * *p* < 0.05 by two-tailed *t*-test.

**Table 2 jcm-12-00612-t002:** Body composition parameters by bioimpedance analysis and cross-sectional areas of the psoas and multifidus muscles.

	Mild/Moderate Disability (n = 29)	Severe Disability (n = 34)	*p* Value
Number of patients			
Male	14	11	
Female	15	23	
BMI (kg/m^2^)			
Male	23.8 ± 2.8	25.3 ± 2.5	0.161
Female	23.4 ± 2.9	25.0 ± 3.5	0.149
Body fat mass (kg)			
Male	10.1 ± 4.8	15.4 ± 6.1	0.014 *
Female	17.4 ± 8.7	21.0 ± 6.0	0.237
Percent body fat (%)			
Male	14.6 ± 6.0	23.2 ± 9.4	0.016 *
Female	27.4 ± 7.4	34.5 ± 6.0	0.004 *
No. of obese patients (%) ^†^			
Male	7 (50.0%)	8 (66.7%)	0.414
Female	5 (31.3%)	16 (73.3%)	0.028 ^‡^
Skeletal muscle mass (kg)			
Male	32.1 ± 3.4	30.4 ± 4.7	0.320
Female	20.8 ± 2.8	21.1 ± 3.2	0.763
Skeletal muscle index (kg/m^2^)			
Male	9.3 ± 1.3	9.6 ± 0.9	0.678
Female	7.3 ± 1.4	7.1 ± 1.0	0.624
CSA of psoas (mm^2^)			
Male	1147.3 ± 119.8	539.9 ± 145.5	<0.001 *
Female	879.4 ± 219.2	619.9 ± 175.1	0.020 *
CSA of multifidus (mm^2^)			
Male	676.0 (540.5–804.1)	398.3 (324.4–510.6)	<0.001 ^§^
Female	575.1 ± 178.0	459.3 ± 93.4	0.146

Data are presented as numbers, means ± standard deviations and median (interquartile range). BMI, body mass index; CSA, cross-sectional area. * *p* < 0.05 by two-tailed *t*-test. ^†^ Obese patients were judged from percent body fat ≥ 32% in women and ≥17% in men. ^‡^ *p* < 0.05 by Chi-square test. ^§^
*p* < 0.05 by Mann–Whitney rank-sum test.

**Table 3 jcm-12-00612-t003:** Epidural block-related profile.

	Mild/Moderate Disability (n = 29)	Severe Disability (n = 34)	*p* Value
NRS (0–10) at first visit	6.0 (5.0–7.8)	7.0 (6.0–8.0)	0.041 *
NRS (0–10) at 4-wks follow-up	3.2 ± 1.3	5.4 ± 2.1	<0.001 ^†^
Pain relief (%) during 4 weeks	45.1 ± 20.0	25.2 ± 4.6	0.002 ^†^
Number of epidural blocks during 4 weeks	2.0 (1.0–2.0)	1.5 (1.0–2.0)	0.155

NRS, Numeric rating scale. * *p* < 0.05 by Mann–Whitney rank-sum test. ^†^ *p* < 0.05 by two-tailed *t*-test.

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
