# Peer review of "The Effects of Body Composition Characteristics on the Functional Disability in Patients with Degenerative Lumbar Spinal Stenosis"

_jcm, 2023, doi:10.3390/jcm12020612_

Round 1
Reviewer 1 Report
Thank you very much for the opportunity of reviewing this study. As one would expect from this group, investigation was carefully conducted. However, this study has some problems.
I wrote the message below to ask you.
I look forward to your reply to my thoughts.
#1 The aim of this study was to determine if an association exists between body composition characteristics and functional disability in ADL of LSS patients, especially to understand body fat parameters could accurately predict it. We further expected that 54 other body composition parameters related to the amount of skeletal muscle also affects it.
Therefore, I think the investigation of effects of epidural steroid injection was not associated with the purpose of this study. Please exclude epidural steroid injection form this study or mention about this in the purpose.
#2 There were no significant difference between mild/moderate and severe groups regarding of radiological severity of LSS in this study. Please discuss about this in the discussion.
#3 The mechanism of association between percent body fat and functional disability in ADL in LSS patients remains unclear. Please mention this in the discussion.
#4 Please show LSS type (cauda equina, radiculopathy, or mixed type).
#5 Were patients with surgical indications included in this study?
#6 Please clarify whether patients took analgesic medication or not. If patients took these medications, please clarify type of agents.
#7 Was the linear regression analysis an age-corrected analysis?
Author Response
Point 1: The aim of this study was to determine if an association exists between body composition characteristics and functional disability in ADL of LSS patients, especially to understand body fat parameters could accurately predict it. We further expected that 54 other body composition parameters related to the amount of skeletal muscle also affects it.
Therefore, I think the investigation of effects of epidural steroid injection was not associated with the purpose of this study. Please exclude epidural steroid injection form this study or mention about this in the purpose.
Response 1: The authors added the mention about epidural steroid injection in line 56-57 in the introduction section.
Point 2: There were no significant difference between mild/moderate and severe groups regarding of radiological severity of LSS in this study. Please discuss about this in the discussion.
Response 2: The authors added the sentences in line 243-245 in the discussion section.
Point 3: The mechanism of association between percent body fat and functional disability in ADL in LSS patients remains unclear. Please mention this in the discussion.
Response 3: The authors added the sentences in line 271-272 in the discussion section.
Point 4: Please show LSS type (cauda equina, radiculopathy, or mixed type).
Response 4: The authors added the words in line 64-65 in the method section.
Point 5: Were patients with surgical indications included in this study?
Response 5: The authors added this to exclusion criteria in line 68-69 in the method section.
Point 6: Please clarify whether patients took analgesic medication or not. If patients took these medications, please clarify type of agents.
Response 6: We added the number of patients who was taking analgesic medications in Table 1.
Point 7: Was the linear regression analysis an age-corrected analysis?
Response 7: The linear regression analysis was age-corrected analysis.

Reviewer 2 Report
We read with great interest this paper, which aimed to ascertain the association between obesity and the severity or ADL of lumbar spinal stenosis(LSS) and concluded thatincreased percent body fat predict potential severe functional disability in ADL in LSS patients..
However, there are several concerns with the acceptance of this paper.
1) Since the association between obesity and metabolic syndrome diseases (diabetes, hypertension, hyperlipidemia) is well known, the breakdown should be examined in this study.
An association between metabolic syndrome diseases (diabetes, hypertension, hyperlipidemia) and lumbar spinal canal stenosis has also been reported by Uesugi.
We are interested in understanding the impact of potentially confounding metabolic diseases.
Uesugi K, Sekiguchi M, Kikuchi S, Konno S. Relationship between lumbar spinal stenosis and lifestyle-related disorders: a cross-sectional multicenter observational study. Spine (Phila Pa 1976). 2013;38(9):E540-E545. doi:10.1097/BRS.0b013e31828a2517
2) The association of obesity with epidural lipomatosis, ligament ossification and DISH is also well known, but these may also be confounding factor.
Ishihara S, Fujita N, Azuma K, et al. Spinal epidural lipomatosis is a previously unrecognized manifestation of metabolic syndrome. Spine J. 2019;19(3):493-500. doi:10.1016/j.spinee.2018.07.022
Okada E, Ishihara S, Azuma K, et al. Metabolic Syndrome is a Predisposing Factor for Diffuse Idiopathic Skeletal Hyperostosis. Neurospine. 2021;18(1):109-116. doi:10.14245/ns.2040350.175
3) Line 247
The authors indicated a relationship between obesity and disc degeneration and osteoarthritis of the spine, but why hasn't these been investigated?
4) Line 289
The author describes the paraspinal muscles, but this is a survey of the psoas major and multifidus muscles, which should be accurately described. In addition, I think a little more discussion of the causal relationship with the psoas major muscle would enhance the significance of this paper.
5) Do improvements in obesity and body composition characteristics correlate with improvements in clinical symptoms and treatment outcomes? I would like to know the literature discussion.
Author Response
Point 1: Since the association between obesity and metabolic syndrome diseases (diabetes, hypertension, hyperlipidemia) is well known, the breakdown should be examined in this study.
An association between metabolic syndrome diseases (diabetes, hypertension, hyperlipidemia) and lumbar spinal canal stenosis has also been reported by Uesugi.
We are interested in understanding the impact of potentially confounding metabolic diseases.
Uesugi K, Sekiguchi M, Kikuchi S, Konno S. Relationship between lumbar spinal stenosis and lifestyle-related disorders: a cross-sectional multicenter observational study. Spine (Phila Pa 1976). 2013;38(9):E540-E545. doi:10.1097/BRS.0b013e31828a2517
Response 1: The authors added the information about metabolic disease in the Table 1.
Point 2: The association of obesity with epidural lipomatosis, ligament ossification and DISH is also well known, but these may also be confounding factor.
Ishihara S, Fujita N, Azuma K, et al. Spinal epidural lipomatosis is a previously unrecognized manifestation of metabolic syndrome. Spine J. 2019;19(3):493-500. doi:10.1016/j.spinee.2018.07.022
Okada E, Ishihara S, Azuma K, et al. Metabolic Syndrome is a Predisposing Factor for Diffuse Idiopathic Skeletal Hyperostosis. Neurospine. 2021;18(1):109-116. doi:10.14245/ns.2040350.175
Response 2: When the authors evaluated MRI images of all participants, there was no those confounding factors. We added those condition to exclusion criteria in line 68-70.
Point 3: Line 247
The authors indicated a relationship between obesity and disc degeneration and osteoarthritis of the spine, but why hasn't these been investigated?
Response 3: Thank you for your thoughtful consideations. Because our study participants were concentrated to degenerative lumbar spinal stenosis, radiologic grading was focused on the LSS itself. We will accept your suggestions and integrated them into a new study design. Thank you.
Point 4: Line 289
The author describes the paraspinal muscles, but this is a survey of the psoas major and multifidus muscles, which should be accurately described. In addition, I think a little more discussion of the causal relationship with the psoas major muscle would enhance the significance of this paper.
Response 4: We modified the words for preventing potential misunderstading. And we will accept your suggestions and integrated them into a new study design. Thank you.
Point 5: Do improvements in obesity and body composition characteristics correlate with improvements in clinical symptoms and treatment outcomes? I would like to know the literature discussion.
Response 5: We added the associated information with references in 306-308.

Round 2
Reviewer 1 Report
I believe that the authors have answered the reviewers' questions appropriately. This manuscript will be worthy of publication.